# Effectiveness of protected areas in conserving tropical forest birds

Victor Cazalis [1✉], Karine Princé[2,3,4], Jean-Baptiste Mihoub[2], Joseph Kelly [1,5], Stuart H. M. Butchart [6,7] & Ana S. L. Rodrigues [1]

Protected areas (PAs) are the cornerstones of global biodiversity conservation efforts, but to fulfil this role they must be effective at conserving the ecosystems and species that occur within their boundaries. Adequate monitoring datasets that allow comparing biodiversity between protected and unprotected sites are lacking in tropical regions. Here we use the largest citizen science biodiversity dataset – eBird – to quantify the extent to which protected areas in eight tropical forest biodiversity hotspots are effective at retaining bird diversity. We find generally positive effects of protection on the diversity of bird species that are forest-dependent, endemic to the hotspots, or threatened or Near Threatened, but not on overall bird species richness. Furthermore, we show that in most of the hotspots examined this benefit is driven by protected areas preventing both forest loss and degradation. Our results provide evidence that, on average, protected areas contribute measurably to conserving bird species in some of the world's most diverse and threatened terrestrial ecosystems.

[1] CEFE, Univ. Montpellier, CNRS, EPHE, IRD, Univ. Paul Valéry Montpellier 3, Montpellier, France. [2] Centre d'Écologie et des Sciences de la Conservation (CESCO), Muséum National d'Histoire Naturelle, Centre National de la Recherche Scientifique, Sorbonne Université, CP 135, 43 rue Buffon, 75005 Paris, France. [3] Université de Lyon, F-69000 Lyon; Université Lyon 1; CNRS, UMR5558, Laboratoire de Biométrie et Biologie Evolutive, F-69622 Villeurbanne, France. [4] Department of Forest and Wildlife Ecology, University of Wisconsin-Madison, Madison, WI 53706, USA. [5] Center for Interdisciplinary Research on Ecology and Sustainability, College of Environmental Studies, National Dong Hwa University, Shoufeng, Hualien 97401, Taiwan. [6] BirdLife International, David Attenborough Building, Pembroke Street, Cambridge CB2 3QZ, UK. [7] Department of Zoology, University of Cambridge, Downing Street, Cambridge CB2 3EJ, UK. ✉email: victor.cazalis@laposte.net

Hopes for halting and reversing the ongoing global biodiversity crisis are largely pinned on protected areas[1,2]. Defined as geographical spaces that are recognised, dedicated, and managed to achieve the long-term conservation of nature[1], they are expected to buffer ecosystems and species populations against some of the most destructive impacts of human activities, particularly those resulting in habitat loss or degradation, or the overexploitation of wildlife. Already covering nearly 15.2% of the global land surface and 7.4% of the oceans[1], signatories to the Convention on Biological Diversity have committed through Aichi Target 11 to expand protected area coverage to 17% and 10%, respectively, by 2020[3], and there are calls to go even further[4]. However, protected areas can only fulfil their intended role if they are effective.

Protected area effectiveness can be assessed through multiple, complementary approaches, for instance, by evaluating whether they cover the diversity of species and ecosystems and the most important sites, or by assessing their management adequacy in terms of staff or resources[1,5]. Here, we focus on effectiveness in terms of biodiversity outcomes: the extent to which the establishment of protected areas makes a difference to the trends and thus ultimately to the condition of the species and ecosystems within their boundaries.

Evaluating outcomes is not straightforward as it requires contrasting current state with a counterfactual, i.e. an alternative scenario of what would have happened if the protected area had not existed[6]. Simply contrasting any protected and unprotected sites would not be an adequate counterfactual analysis, because it would conflate implementation effects (the difference protected areas have made) with location biases (differences between protected and unprotected sites prior to protected areas implementation)[5,6]. Such location biases are inevitable because protected areas tend to be designated in regions of little economic interest (i.e. greater remoteness, higher altitudes, and lower agricultural suitability[6,7]), which are less likely to have suffered from human pressure both before and after protection. These differences can be statistically controlled for in counterfactual analyses of protected area effectiveness[8–10], but this requires large datasets on the spatial distribution of the biodiversity features of interest across many protected and unprotected sites.

Nowhere are effective protected areas more essential than in tropical regions, which host a disproportionately large share of the world's biodiversity[11] and face rapid habitat loss[11] and degradation[12,13], both major threats to biodiversity[13–15]. Yet, evaluating protected area effectiveness in these regions is particularly challenging, given that the detailed biodiversity datasets required for counterfactual analyses are typically unavailable[16]. Among the few analyses investigating biodiversity outcomes of tropical protected areas, most focused on protected area effects on land cover, finding that they mitigate both forest loss and forest degradation[10,17–19]. While such analyses are possible using ex situ remote sensing data, investigating effectiveness in terms of species outcomes requires data collected in situ. Two global meta-analyses reviewed local-scale studies that had contrasted protected versus unprotected sites in terms of species diversity[9,20]. Both uncovered positive effects at the global scale, but—worryingly—weaker or mixed results in tropical regions, contrasting with reported positive effects of protected areas at reducing forest loss and degradation.

In this study, we investigate the effectiveness of protected areas in eight tropical forest biodiversity hotspots across three continents (Fig. 1), which are the epicentres of the ongoing biodiversity crisis and therefore regions where effective conservation efforts are the most urgent[11,21,22]. For this purpose, we take advantage of eBird[23,24], the world's largest citizen science

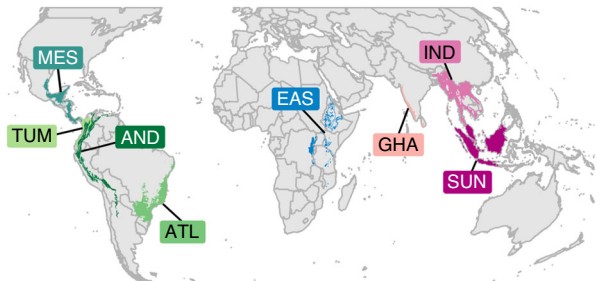

**Fig. 1 Regions covered by the present study (i.e. intersection between eight biodiversity hotspots and the "tropical and subtropical moist broadleaf forests" biome).** Acronyms as in Figs. 3 and 4: ATL (Atlantic Forest, $N = 6760$ checklists), AND (Tropical Andes, $N = 17,758$), TUM (Tumbes-Chocó-Magdalena, $N = 1188$), MES (Mesoamerica, $N = 32,784$), EAS (Eastern Afromontane, $N = 1097$), GHA (Western Ghats and Sri Lanka, $N = 2646$), IND (Indo-Burma, $N = 2996$), and SUN (Sundaland, $N = 1548$). More details in Supplementary Fig. 1.

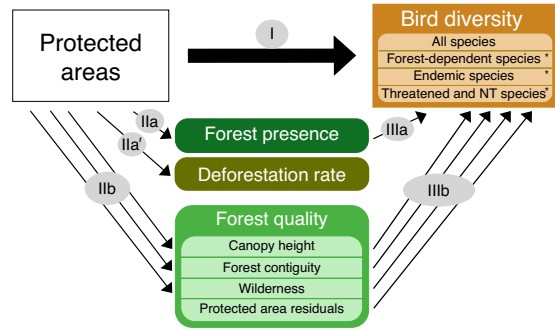

**Fig. 2 Framework of the analyses performed to investigate the effectiveness of protected areas at retaining bird diversity.** Analysis I: effect of protected areas on bird diversity measured through four indices of bird species richness (all species, forest-dependent species, endemic species, threatened, and near threatened species). The asterisk indicates species of conservation concern. Analysis II: effects of protected areas on forest presence (IIa), local deforestation rates (IIa′), and on three measures of forest quality (canopy height, forest contiguity, and wilderness; IIb). Analysis III: effects of forest presence (IIIa), and of each of the three measures of forest quality and of the residual effect of protected areas (IIIb) on bird diversity.

programme that provides fine-scale occurrence data of bird species, and we quantify protected areas outcomes for bird species diversity using a counterfactual analysis that controls for location biases. For each hotspot, we apply a set of three distinct but interrelated statistical analyses to investigate the effectiveness of protected areas at retaining bird diversity, and to shed light on the underlying mechanisms (Fig. 2). First (analysis I), we show that protected areas do not retain more species than unprotected counterfactuals, but that they do retain more species of greater conservation concern, namely specialists (here, forest-dependent species), species with narrow ranges (i.e. endemic to the hotspot), and species classified as threatened (Critically Endangered, Endangered, or Vulnerable) or Near Threatened in the IUCN Red List[25]. We consider two potential mechanisms through which protected areas can potentially affect bird diversity: by retaining forest presence (i.e. mitigating forest loss), and by maintaining forest quality (i.e. mitigating forest degradation). We test these mechanisms in two complementary analyses (Fig. 2). One (analysis II) shows that protected areas have a positive effect on forest presence (IIa), by mitigating local deforestation rates (2000–2019;

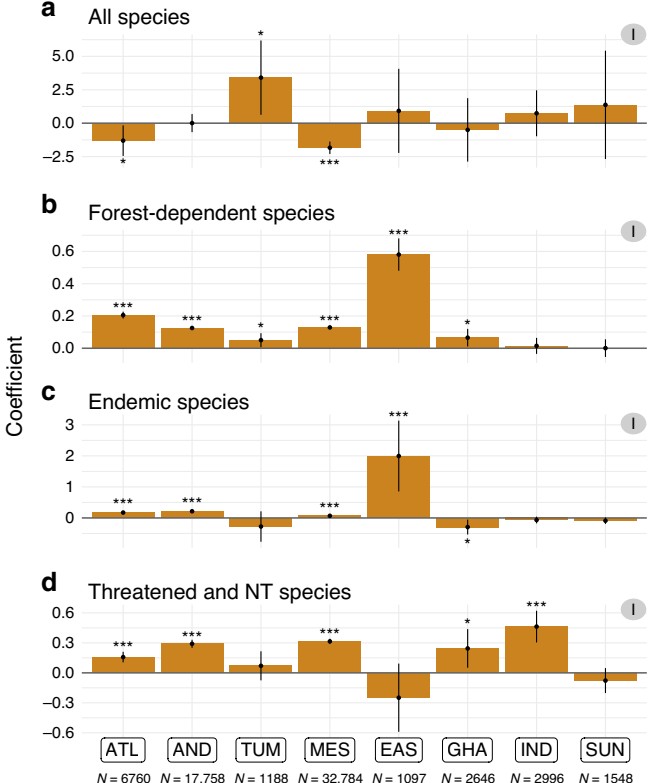

**Fig. 3 Effect of protected areas on bird diversity per hotspot, for four bird diversity indices (analysis I). a** Overall species richness. **b** Forest-dependent species richness. **c** Endemic species richness. **d** Richness in threatened and Near Tthreatened species. Coefficients correspond to the estimates of GAM models; significance is given by the $P$ value (***<0.001 < **<0.01 < *<0.05, see details in Supplementary Table 3) and the 95% confidence interval around GAM coefficients (vertical error bars). Hotspots: ATL (Atlantic Forest), AND (Tropical Andes), TUM (Tumbes-Chocó-Magdalena), MES (Mesoamerica), EAS (Eastern Afromontane), GHA (Western Ghats and Sri Lanka), IND (Indo-Burma), and SUN (Sundaland). Number of checklists per hotspot is specified below hotspots names; more detailed results in Supplementary Table 1.

IIa′), but also on forest quality (IIb; measured by canopy height, forest contiguity—the opposite of fragmentation—, and wilderness—the opposite of the human footprint index[26]). The other (analysis III) shows the positive effects of either forest presence (IIIa) or forest quality (IIIb) on bird species of conservation concern.

### Results

**Protected areas mitigate the replacement of species of concern**. We found no consistent evidence across hotspots of an effect of protected areas on overall richness in bird species (analysis I). Indeed, we obtained non-significant results for five out of the eight hotspots tested, significant negative effects for two, and a significant positive effect for a single one (Fig. 3a). Given that species richness is an intuitive and widely used measure of biodiversity[27], these results may appear worrying, by suggesting that protected areas do not prevent local biodiversity loss. In fact, they agree with a wealth of previous evidence that overall species richness is not a suitable indicator of local biodiversity impact, as species that go locally extinct due to ecosystem alteration can be replaced by others—often of lower conservation concern—with no or little impact on overall species

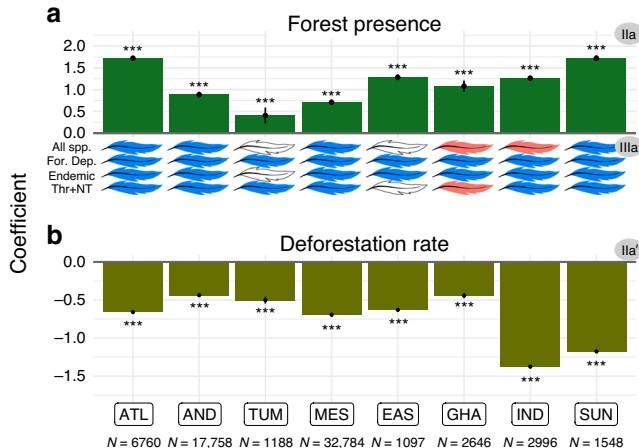

**Fig. 4 Effects of protected areas on forest cover, and effects of forest cover on bird diversity, per hotspot.** Bars: effects of protected areas on forest presence (**a** analysis IIa) or on deforestation rates between 2000 and 2019 (**b** analysis IIa′); coefficients correspond to the estimates of GAM models; significance given by $P$ value (***<0.001 < **<0.01 < *<0.05), and 95% confidence interval around GAM coefficients (vertical error bars). Feathers: colour represents the effect sign (blue: positive; red: negative; white: non-significant [$P$ value > 0.05]) of forest presence on each of the bird diversity variables (All spp., overall species richness; For. Dep., richness in forest-dependent species; Endemic, richness in endemic species; Thr + NT, richness in threatened and Near Threatened species). Hotspots: ATL (Atlantic Forest), AND (Tropical Andes), TUM (Tumbes-Chocó-Magdalena), MES (Mesoamerica), EAS (Eastern Afromontane), GHA (Western Ghats and Sri Lanka), IND (Indo-Burma), and SUN (Sundaland). Number of checklists per hotspot is specified below hotspots names; more detailed results in Supplementary Table 1 and Supplementary Fig. 3.

richness[27,28]. Accordingly, we also found that neither forest presence (Fig. 4a; Supplementary Fig. 3a; analysis IIIa) nor forest quality (Fig. 5a–c; Supplementary Fig. 3b–d; analysis IIIb) had a consistent positive effect on overall species richness, indicating that this diversity measure is rather insensitive to habitat loss and degradation, at least at the temporal and spatial scales considered in this study. Species richness is also known to temporarily increase at intermediate levels of disturbance[29], perhaps explaining the few observed negative effects of protection (Fig. 3a), forest presence (Fig. 4a), and forest quality (Fig. 5) on overall species richness.

Whereas we found no effect of protected areas on overall species richness, our results indicate that protected areas are effective at retaining the three types of species of conservation concern that were analysed: forest dependent (i.e. specialists), endemics to each hotspot (i.e. narrow ranged), and threatened or Near Threatened (i.e. at greater risk of extinction). Indeed, controlling for overall richness, we find for each of these three groups significant positive effects of protected areas across hotspots (Fig. 3b–d; analysis I), particularly for forest-dependent species (in 6 out of 8 hotspots; with protected sites on average 17.8% richer in forest-dependent species than comparable unprotected sites; Fig. 3b), but also for endemic species (4/8; 77.6%; Fig. 3c) and threatened and Near Threatened species (5/8; 19.0%; Fig. 3d; Supplementary Table 1). The consistency in these results may be derived from the partial overlap between the species classified in these three classes of conservation concern (Supplementary Fig. 2). Overall, our results indicate that protected areas are effective at avoiding the replacement of species of conservation concern (specialists, with small ranges,

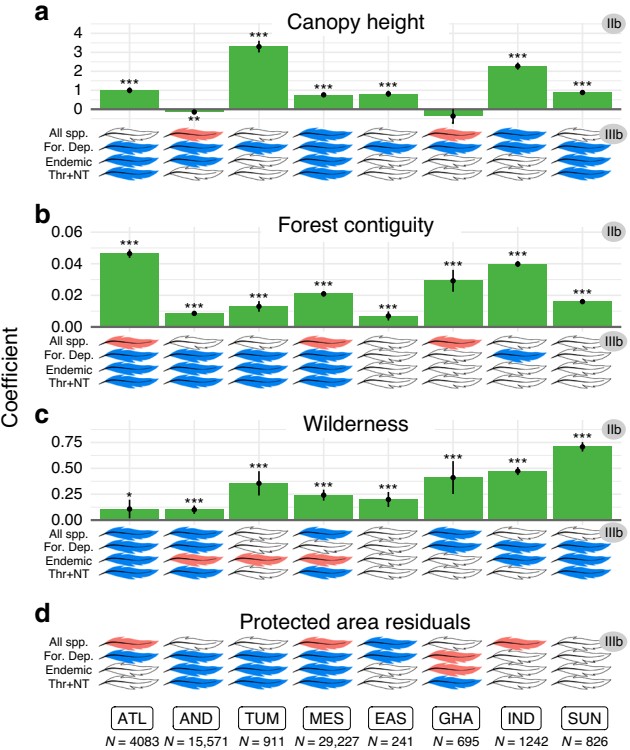

**Fig. 5 Effects of protected areas on forest quality, and effects of forest quality on bird diversity, per hotspot.** Bars: effects of protected areas on forest quality (**a** canopy height; **b** forest contiguity; **c** wilderness; **d** protected area residuals; analysis IIb); coefficients correspond to the estimates of GAM models; significance given by $P$ value (***<0.001< **<0.01 < *<0.05), and 95% confidence interval around GAM coefficients (vertical error bars). Feathers: colour represents the effect sign (blue: positive; red: negative; white: non-significant [$P$ value > 0.05]) of each habitat quality variable on each of the bird diversity variables (All spp., overall species richness; For. Dep., richness in forest-dependent species; Endemic, richness in endemic species; Thr + NT, richness in threatened and Near Threatened species). Hotspots: ATL (Atlantic Forest), AND (Tropical Andes), TUM (Tumbes-Chocó-Magdalena), MES (Mesoamerica), EAS (Eastern Afromontane), GHA (Western Ghats and Sri Lanka), and IND (Indo-Burma), SUN (Sundaland). Number of checklists per hotspot is specified below hotspots names (only forest checklists in this analysis); more detailed results in Supplementary Table 1 and Supplementary Fig. 3.

and at higher risk of extinction) by more widespread and generalist species.

Our results corroborate studies in temperate regions that found that protected areas do not protect all species and thus do not always affect species richness[8,30]. However, they contrast with what was known in tropical regions from two previous global-scale studies of protected area effectiveness, based on the meta-analysis of local-scale studies contrasting protected versus unprotected sites. One of these studies found higher species richness and abundances within protected areas in Africa and Asia but not in South America[20]; the other found higher overall richness within protected areas, but no significant effects on the richness in rare and endemic species, including in the tropics[9]. Nonetheless, the present study provides a stronger test of protected area effectiveness in tropical forests by focusing specifically on these biomes, using more comparable data (as emerging from a single, coherent dataset), and by exploring the underlying mechanisms of habitat loss and degradation. We have controlled for confounding variables in order to separate the

implementation effects of protected areas from potential location biases (see Supplementary Methods 4d), and have also found that older protected areas tend to be more effective (in terms of conserving bird diversity; analysis I), consistently with a cumulative implementation effect of protected areas (Supplementary Methods 4e, Supplementary Fig. 18).

**Protected areas retain species of concern by mitigating forest loss.** Our results suggest that protected areas effectiveness at retaining species of concern is mainly driven by their effectiveness at mitigating forest loss. First, we found significant positive effects of protection on forest presence across all hotspots analysed, with a protected site having on average a 17.8% higher probability of being forested than a non-protected counterfactual (Fig. 4a; analysis IIa; Supplementary Table 1). Second, we confirmed the temporal effect of protected areas on forests, showing across all hotspots that deforestation rates within protected areas were lower (on average 46.7% so) than in non-protected counterfactuals (Fig. 4b; analysis IIa'; Supplementary Table 1). These results confirm and extend previous works showing positive effects of protected areas at reducing rates of tropical deforestation[10,17,31]. Third, we found that forested sites have higher richness in forest-dependent bird species than comparable non-forested sites (across 8/8 hotspots; on average 74.9% more species), as well as in endemic species (7/8; 250.0%) and in threatened and Near Threatened species (6/8; 122.1%; analysis IIIa; Fig. 4a; Supplementary Table 1; Supplementary Fig. 3a), in accordance with the well-known devastating impact of deforestation on biodiversity[11,13,15]. Particularly in line with our results, Rutt et al.[32], have highlighted the replacement of forest-dependent bird species by generalist species following experimental deforestation in the Amazon.

**Protected areas retain species of concern by mitigating forest degradation.** Our results further indicate that the added value of protected areas towards the conservation of species of concern also comes from their mitigation of forest degradation. Firstly, we found a generally positive effect of protection on forest quality (analysis IIb), as measured through each of three variables: canopy height (6/8 hotspots; on average 4.8% higher in protected than in counterfactual forested non-protected sites; Fig. 5a), forest contiguity (8/8; 2.6% higher; Fig. 5b), and wilderness (8/8; 5.7% higher; Fig. 5c; Supplementary Table 1). The last is the reciprocal result of two recent studies showing lower levels of human pressure within protected areas when compared with appropriate counterfactuals in tropical forests[18,19].

Secondly, our results suggest that each of these three variables of habitat quality increases richness in species of concern. Indeed, we show a positive effect of canopy height (in 8/8 hotspots for richness in forest-dependent species; 4/8 for endemic species; 3/8 for threatened and Near Threatened species; Fig. 5a; analysis IIIb; Supplementary Fig. 3b), of forest contiguity (in 5/8 hotspots for forest-dependent; 4/8 for endemic and for threatened and Near Threatened species; Fig. 5b; analysis IIIb; Supplementary Fig. 3c) and of wilderness (in 5/8 for forest-dependent species; 3/8 for endemics [but also 3/8 negative]; 4/8 for threatened and Near Threatened species; Fig. 5c; analysis IIIb; Supplementary Fig. 3d). Finally, even after controlling for canopy height, contiguity, and wilderness, we found that among forested protected areas there are generally positive residual effects of protection itself on forest-dependent species (positive in 5/8 hotspots; but 1/8 negative), on endemics (3/8 positive; but 1/8 negative), and on threatened and Near Threatened species (4/8 positive) (Fig. 5d; Supplementary Fig. 3e). This indicates that the positive effect of protection in mitigating forest degradation goes beyond the three habitat

quality variables we have considered, perhaps reflecting reductions of other pressures such as hunting, selective logging, or invasive species[33,34].

**Stronger evidence for the effectiveness of South American protected areas.** We found substantial variability across hotspots in the effects of protected areas on the diversity of species of concern. Indeed, the most consistent picture emerges for three of the American hotspots—Atlantic Forest (ATL), Tropical Andes (AND), and Mesoamerica—for which we found consistently significant positive effects of protection on the three groups of species of concern (Fig. 3), with both forest presence (Fig. 4) and forest quality (Fig. 5) playing seemingly important roles. Results were more mixed for the other hotspots. We found significant effects of protection on the diversity of forest-dependent species for the Tumbes-Chocó-Magdalena hotspot (TUM), of forest-dependent and endemic species for the Eastern Afromontane hotspot (EAS), of forest-dependent species and threatened and near threatened—as well as a negative effect on endemic species—for the Western Ghats and Sri Lanka (GHA), of species of concern for Indo-Burma (IND), and no significant effects on species of conservation concern for Sundaland (SUN). This may reflect variation in the effectiveness of protected area implementation across the world, or simply differences in statistical power. Indeed, the three American hotspots with the strongest signal of effectiveness are those with the most data (6760–32,784 checklists, contrasted with 1097–2996 for the other hotspots; Supplementary Fig. 1; see Supplementary Discussion for further discussion on heterogeneity in the results and Supplementary Table 3) and for which data are the most homogeneously distributed with hotspots (Supplementary Figs. 4 and 5).

## Discussion

We provide evidence for the effectiveness of protected areas as biodiversity conservation tools across eight global biodiversity hotspots, covering some of the planet's most diverse and threatened terrestrial ecosystems[21]. We used a counterfactual analysis that controls for location biases in the establishment of protected areas. Although these controls are necessarily imperfect (see Supplementary Methods 4d), we aimed to isolate as much as possible the effects of implementation itself, i.e. the added value of protection. We found that this value does not lie in preventing declines in overall local species richness, but in avoiding the replacement of species that are most in need of conservation efforts: the forest specialists that are most at risk from forest loss or degradation; the endemic species that make each hotspot globally irreplaceable; and threatened or Near Threatened that are at higher risk of global extinction.

Our results contribute to the body of evidence supporting the effectiveness of protected areas at avoiding forest loss[10,17], focusing here on tropical forests of biodiversity hotspots. Furthermore, they indicate that this is the main mechanism through which protection has a positive effect on retaining bird species of concern. In addition, we provide evidence that it is not the only mechanism, with protection also having a significant effect on bird diversity by mitigating forest degradation, as measured through canopy height, fragmentation and wilderness levels. Finally, we found evidence for a residual effect of protection (once controlling for the effects on forest presence and quality) that may reflect management measures of other pressures such as hunting, small-scale logging, or invasive species.

In this study, we found that protected areas are effective in the sense that they perform better than comparable unprotected sites. We have, however, not demonstrated that they are sufficiently effective to halt habitat loss and degradation (which previous studies found to be ongoing and sometimes increasing within protected areas[17–19]) nor that they halt population declines (which are still ongoing within many protected areas[35–37]). Furthermore, our analysis does not address whether protected areas are sufficient in terms of their extent or representativeness (while previous studies attest that they are not[38,39]). Nonetheless, our results indicate that protected areas are already making a measurable difference in terms of biodiversity conservation in several regions of the world where the conservation stakes are the highest. In this year when Aichi Targets are due to be reached[3], yet some governments are announcing protected areas degazettement and downsizing[40], our results support the key role of protected areas as global biodiversity conservation tools. We therefore join calls for the strategic expansion of the global protected areas estate and increased investment to ensure that they are effectively managed[41–43].

## Methods

**Study areas: biodiversity hotspots.** We focused on eight biodiversity hotspots[21]: those with at least 25% of their extent within the "tropical and subtropical moist broadleaf forests" biome[44] and for which we obtained at least 1000 checklists from eBird (after applying the data selection procedure described below): Atlantic Forest, Tropical Andes, Tumbes-Chocó-Magdalena, and Mesoamerica (Americas); Eastern Afromontane (Africa); Western Ghats and Sri Lanka, Indo-Burma and Sundaland (Asia). Within each hotspot, we analysed only areas overlapping the "tropical and subtropical moist broadleaf forests" biome[44] (Fig. 1, Supplementary Figs. 1, 4, and 5), assumed to have been originally forested (see Supplementary Methods 4d).

**Data selection: eBird checklists.** We obtained bird sightings from the eBird citizen science database[23]. The reporting system is based on checklists, whereby the observer provides: list of birds detected; GPS location; sampling effort (whether or not all detected species are reported; sampling duration; sampling protocol, e.g., stationary point, travel, and banding; and distance travelled in case of travelling protocol); starting time of the sampling event; and number of observers.

We used the eBird dataset released in December 2018[45], focusing on records from 2005 to 2018, as data collected prior to 2005 were too scarce for analysis. We filtered this dataset to obtain high-quality checklists comparable in protocol and effort: we selected complete checklists only (i.e. in which observers explicitly declare having reported all bird species detected and identified); following either the "stationary points" or the "travelling counts" protocol; with durations of continuous observation of 0.5–10 h; with observers travelling distances during the checklist < 5 km; only from experienced observers (≥10 checklists; ≥30 species per checklist on average; ≥100 different species in total); and removing potential duplicates (checklists made on the same day at the same place). We applied some taxonomical transformation to eBird data in order to fit with BirdLife International taxonomy (Supplementary Methods 1f).

After data filtering (more details in Supplementary Methods 1), we obtained the final dataset used in the analyses, consisting of 66,777 checklists, covering 5467 species, from 6838 observers, in eight hotspots (Supplementary Figs. 4 and 5; Supplementary Table 2).

**Site characteristics.** Our analyses included two types of sites: checklist sites, corresponding to the coordinates of each eBird checklist analysed (used in analyses I and III); and background sites, corresponding to the centre points of a regular grid of $2 \times 2$ km covering the whole area of each hotspot evenly (used in analysis II). We characterised each site according to six characteristics—calculated in a 1-km radius buffer around its coordinates—two binary and four continuous: protected (if coordinates fall within a protected area[46]; Supplementary Fig. 6) versus non-protected; forest (if >60% of the 1-km buffer around the point is forested[47]) versus non-forest (<10% forested; sites with intermediate forest cover were removed from analyses); altitude[48]; agricultural suitability[49]; remoteness[50]; and the proportion of forest loss between 2000 and 2019[51]. In addition, we classified each forest site according to three continuous variables: canopy height[52]; forest contiguity (proportion of forest cover[47], 0.6–1); and wilderness level (opposite of human footprint[53]).

Finally, checklist sites were also characterised according to four measures of local bird diversity: overall species richness (total number of species detected in the checklist); richness in forest-dependent species (high or medium dependency on forest habitats[25]); richness in endemic species (at least 90% of their global distribution within a hotspot[54]); and richness in species of concern (classified as Near Threatened or threatened, i.e. Vulnerable, Endangered, or Critically Endangered[25]; more details in Supplementary Methods 2).

**Index of observer expertise.** Heterogeneity in observers' birding skills, behaviours, and equipment increases data variability and potentially introduces biases

to the analyses[55,56]. Heterogeneity is particularly high in citizen science datasets like eBird, where volunteers range from those only familiar with a few common local birds to experienced observers capable of detecting rare and cryptic species. As stated above, we only included checklists from relatively experienced observers. To account for the remaining variability in observer expertise, we calculated an observer expertise score (used as an explanatory variable in the statistical analyses), adapted from Kelling et al.[57] and from Johnston et al.[56], and calculated separately for each continent. It estimates the variation in the number of species that observers are predicted to detect in similar conditions. To do so, we first ran a mixed general additive model (function *gamm* from "mgcv" R package[58]) modelling species richness of checklists against potential confounding variables that are expected to affect either the number of species detected (sampling *protocol*; *n.observers* number of observers; *duration* of sampling; *time* of the day) or the true species richness (*lat* latitude; *lon* longitude; and Julian *day*), adding *observer* as a random effect

$$\text{gamm}(\text{richness} \sim \text{protocol} + \text{n.observers} + \text{s(duration)} + \text{s(time)}$$
$$+ \text{te(lon, lat, day)} + \text{random} = \text{list(observer} \sim 1)). \quad (1)$$

The notation s() indicates that the variable was used as a smoothed term; and te() indicates that the variables have been used as interacting smooth terms, allowing here species richness to vary spatially during the year.

After fitting this model to each continental data subset, we used it to predict the logarithm of species richness that each observer would report for a fictive stationary point with all variables fixed to their median values. This resulted in an observer expertise score that we then assigned to all checklists; assigning the observer score of the observer with the highest expertise score in cases of multiple observers. This index ranged from 2.2 to 4.3 in Africa, from 2.3 to 4.4 in the Americas, and from 2.8 to 4.5 in Asia (more details in Supplementary Methods 3).

**Statistical analyses of protected area effectiveness**. We investigated protected area effectiveness at retaining bird diversity through a set of three connected statistical analyses (Fig. 2), undertaken separately for each hotspot, using GAM models[58]. The first analysis (I) directly estimated the effects of protection on bird diversity while the two others (II and III) investigated the underlying mechanisms to explain the results of the first analysis.

Analysis I quantifies the effect of protected areas on bird diversity through models contrasting bird diversity of checklist sites between protected versus unprotected sites, while controlling for protected area location biases (and other potential confounding factors)

$$\text{I: Bird\_Diversity} \sim \text{protection} + \text{location\_biases} + \text{control.} \quad (2)$$

Analysis II quantifies the effectiveness of protected areas at mitigating forest loss and forest degradation, through models controlling for location biases and spatial autocorrelation. To measure the effects of protection on forest loss (IIa), we built logistic models contrasting protected versus unprotected background sites in their probability of being forested with land cover data

$$\text{IIa: Forest\_presence} \sim \text{protection} + \text{location\_biases} + \text{te(lon, lat).} \quad (3)$$

We have also run an analysis IIa′ comparing forest loss rates (log transformed to fit normal distribution) between protected and unprotected sites

$$\text{IIa}': \log(0.001 + \text{Forest\_loss}) \sim \text{protection} + \text{location\_biases} + \text{te(lon, lat).} \quad (4)$$

To measure the effects of protected areas on forest degradation (IIb), we built Gaussian models contrasting protected versus unprotected background forested sites in terms of forest quality (canopy height, forest contiguity, or wilderness)

$$\text{IIb: Forest\_quality} \sim \text{protection} + \text{location\_biases} + \text{te(lon, lat).} \quad (5)$$

Analysis III quantifies the effects of forest presence (IIIa) or of forest quality (IIIb) on bird diversity, while controlling for potential confounding factors. In IIIa, we built models contrasting bird diversity in forest versus non-forest checklist sites

$$\text{IIIa: Bird\_Diversity} \sim \text{Forest\_presence} + \text{control.} \quad (6)$$

In IIIb, we modelled local bird diversity of forested sites against the three forest quality variables, as well as protected status in order to capture other aspects of forest quality that could be increased within protected areas (e.g. enforcement of hunting regulations; what we call protected area residuals)

$$\text{IIIb: Bird\_Diversity} \sim \text{scale(canopy)} + \text{scale(contiguity)}$$
$$+ \text{scale(wilderness)} + \text{protection} + \text{control.} \quad (7)$$

In analyses I and III, the response variable *Bird_Diversity* is one of the four metrics of local bird diversity. We assumed Gaussian distribution for the overall richness, and a negative binomial distribution for the richness in forest-dependent species, endemic species and threatened and Near Threatened species.

In analysis II, the response variable is either the binary *Forest_presence* (site forested or not) or each of three measures of *Forest_quality* (canopy height, forest contiguity, or wilderness).

The term *location_bias* in analyses I and II corresponds to $s(altitude) + s(remoteness) + s(agricultural\_suitability)$, supplemented by a control for spatial

autocorrelation in analysis II with the term $+ te(lon, lat)$. It controls for potential biases in protected area location in relation to altitude, remoteness, and agricultural suitability[6,7] (Supplementary Figs. 7–9).

In analyses I and III, we controlled for other potential confounding factors that could affect the bird diversity reported in a checklist (Supplementary Figs. 10–17). In particular, we controlled for: heterogeneity in sampling effort (sampling *duration*; observer *expertise*; number of observers: *n.observers*); temporal effects (*year* to account for possible trends; *day* to account for season); and spatial heterogeneity (*lat* latitude, *lon* longitude). *Lon*, *lat*, and *day* were used as interacting smooth terms, enabling bird diversity variables to vary spatially across seasons (see Supplementary Methods 2 and Supplementary Figs. 10–17). The term control was thus

$$s(\text{duration, k} = 4) + s(\text{expertise, k} = 4) + s(\text{n.observers, k} = 4)$$
$$+ s(\text{year, k} = 4) + te(\text{day, lat, lon}).$$

When the response variable was richness in forest-dependent species, in endemic species or in threatened and Near Threatened species, we also controlled for overall species richness, thus using as control term

$$\log(\text{overall\_richness}) + s(\text{duration, k} = 4) + s(\text{expertise, k} = 4)$$
$$+ s(\text{n.observers, k} = 4) + s(\text{year, k} = 4) + te(\text{day, lat, lon}).$$

In analysis I, altitude is already controlled for under the location_bias term; in analysis III, the control term also includes a term controlling for it: $s(altitude, k = 6)$.

**Reporting summary**. Further information on research design is available in the Nature Research Reporting Summary linked to this article.

## Data availability

The data that support the findings of this study are all freely available for research purposes, and can either be directly downloaded from the respective websites or are available on demand.

Bird species checklists (eBird v. 12/2018), http://ebird.org.

Bird species distribution maps (version V7.0), http://datazone.birdlife.org (available on demand).

Bird species IUCN Red List status (IUCN Red List of Threatened Species, version 2017.1), http://iucnredlist.org.

Protected area polygons (World Database on Protected Areas, v. 10/2018), https://www.protectedplanet.net/.

Elevation data (GLOBE Digital Elevation Model, v.1, 1999), https://www.ngdc.noaa.gov/mgg/topo/gltiles.html.

Biodiversity hotspots, https://www.cepf.net/our-work/biodiversity-hotspots/hotspots-defined.

Biome boundaries, https://www.worldwildlife.org/publications/terrestrial-ecoregions-of-the-world;

Forest cover (Climate Change Initiative Land cover Data Project Map from 2015 v.2.0.7) http://maps.elie.ucl.ac.be/CCI/viewer/index.php.

Agricultural suitability 1981–2010 (version 1.0), https://www.ufz.de/glues/.

Remoteness (Accessibility to Cities, version 1.0), https://malariaatlas.org/research-project/accessibility_to_cities/.

Canopy height (version 1.0), https://landscape.jpl.nasa.gov/.

Human footprint (version 1.0), https://doi.org/10.5061/dryad.052q5.

Forest loss (2000–2019, version 2019-v1.7), http://earthenginepartners.appspot.com/science-2013-global-forest.

## Code availability

All R scripts of analyses used in main text are publicly available at: https://doi.org/10.5281/zenodo.3952401.

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

## Acknowledgements

We are grateful to the numerous observers who contributed records to eBird, and to the eBird team at Cornell Lab of Ornithology for creating and managing this database. J.K. was funded by the European Union's Horizon 2020 research and innovation programme under the Marie Skłodowska-Curie grant agreement No 766417. This communication reflects only the authors' view and the Research Executive Agency of the European Union is not responsible for any use that may be made of the information it contains. K.P. was funded by the French National Research Agency under the grant agreement No ANR-17-CE04-0012.

## Author contributions

V.C. and A.S.L.R. conceived the study, in discussion with K.P. and J.-B.M.; V.C. carried out the statistical analyses, with the support of K.P. and J.-B.M.; V.C. wrote the first draft, supported by A.S.L.R.; J.K and S.H.M.B. helped interpreting the results and all authors commented on and revised the manuscript.

## Competing interests

The authors declare no competing interests.
