## [Peer Review File · Nature Communications]

Reviewers' Comments:

Reviewer #1:

Remarks to the Author:

Thank you very much for the opportunity to review manuscript 252244_0, "Effectiveness of protected areas in conserving tropical forest birds". This is a great and timely paper, providing much needed insights into the effectiveness of protected areas. I was impressed by the authors effort included in this study and would also like to commend the authors effort on both the paper and the supplementary information describing their methodology. It clearly shows that a lot of effort went into this paper and making sure its accessible to readers. Very well done!

I think this paper represents a valuable contribution to the literature and would fit very well with the journal. All data sets used and analytical descriptions make sense to me and the conclusions are well justified given the approach taken and data used. The paper flows logically and the text is well complemented by the figures included in main text. Supplementary text, figures and tables are well done as well and are of great use to readers that are interested in additional details to the main text.

When reading the manuscript, I thought about possible explanations of varying patterns across regions and a potential link to statistical power/sample size. It is my view that the authors have accounted for this as much as possible and in my opinion the mention of this as a caveat in the main text is sufficient to point this out (line 244 on). Reviewing the paper, this was the only potential issue that I could see, but the authors have proactively addressed this, which helps the transparency of this study. Readers can draw their own conclusions, but for me, having worked with eBird data extensively, the authors have made every effort to account for bird data issues and I think this is as far as one can realistically go in trying to explain differences between and within regions.

Two minor comments:

1. It would have been really helpful in the review process to have a look at the R code as well. I know that the authors have included the following code availability statement, but I think it would make sense for reviewer to be able to reproduce the analysis during their review: All R scripts will be deposited on an open repository after revision.

2. The acknowledgements and author contributions sections should be removed for blinding the manuscript.

Best regards,
Richard Schuster

Research Associate, Carleton University
Email: richard.schuster@glel.carleton.ca

Reviewer #2:

Remarks to the Author:

I found the MS entitled "Effectiveness of protected areas in conserving tropical forest birds" to be highly interesting, important, innovative, and well written. I have but few suggestions that could potentially help cement the authors claims regarding the effectiveness of PAs in protecting vulnerable

bird species.

Major comments

The authors claim that PAs were established in more remote, and less desirable locations to begin with. However, there is also a possibility that these regions were specifically chosen to protect at risk (bird) species. I think that this option should at least be raised. Furthermore, there could potentially be some more test to try to disentangle this chicken and egg conundrum.

First, I think that the year PAs were established should be used as another predictor in some of the tests – to see if PAs established earlier are doing better than those more recently established. These data should be available from the UNEP-WCMC database.

Furthermore, beyond the metrics of forest quality explored here, I would suggest looking at the Hansen forest cover-change layer (<https://earthenginepartners.appspot.com/science-2013-global-forest>) which gives nearly 20 years spatial data regarding forest loss. There are similar time-series derived spatial datasets spanning even longer periods. Here too the dynamics of the forest cover/loss are important for linking longer-term biodiversity phenomena.

Put together, placing your results within such an (also) temporal perspective should give them even stronger foundations.

Minor comments

Could you provide in the supp. the power of your different tests for different groups of species, or regions. This could help compare significant / non-significant results across your many tests with different sample sizes.

Could you please also provide a supp. with the lists of species belonging to different categories in different regions, this should help reproducibility of your results, and further explorations of them.

Please note that the Simrad et al. 2011 data is limited to a max canopy height of 40m. I suggest at least mentioning this.

Reviewer #1 (Remarks to the Author):

Thank you very much for the opportunity to review manuscript 252244_0, "Effectiveness of protected areas in conserving tropical forest birds". This is a great and timely paper, providing much needed insights into the effectiveness of protected areas. I was impressed by the authors effort included in this study and would also like to commend the authors effort on both the paper and the supplementary information describing their methodology. It clearly shows that a lot of effort went into this paper and making sure its accessible to readers. Very well done!

I think this paper represents a valuable contribution to the literature and would fit very well with the journal. All data sets used and analytical descriptions make sense to me and the conclusions are well justified given the approach taken and data used. The paper flows logically and the text is well complemented by the figures included in main text. Supplementary text, figures and tables are well done as well and are of great use to readers that are interested in additional details to the main text.
-> We genuinely thank Reviewer 1 for his positive comments.

When reading the manuscript, I thought about possible explanations of varying patterns across regions and a potential link to statistical power/sample size. It is my view that the authors have accounted for this as much as possible and in my opinion the mention of this as a caveat in the main text is sufficient to point this out (line 244 on). Reviewing the paper, this was the only potential issue that I could see, but the authors have proactively addressed this, which helps the transparency of this study. Readers can draw their own conclusions, but for me, having worked with eBird data extensively, the authors have made every effort to account for bird data issues and I think this is as far as one can realistically go in trying to explain differences between and within regions.
-> Indeed, we too spent some time thinking about explanations for this variation across regions. We did not have much space to discuss it in the main text, so we explored more deeply in the Supplementary Discussion (especially whether heterogeneity in the results comes from sampling effort differences, ecological differences, or conservation differences). To make sure the reader does not miss this additional discussion, we now point to it explicitly in the main text ("see Supplementary Discussion for further discussion on heterogeneity in the results").

Two minor comments:

1. It would have been really helpful in the review process to have a look at the R code as well. I know that the authors have included the following code availability statement, but I think it would make sense for reviewer to be able to reproduce the analysis during their review: All R scripts will be deposited on an open repository after revision.

-> Sincere apologies for not having provided the scripts earlier. They are now attached to this submission.

2. The acknowledgements and author contributions sections should be removed for blinding the manuscript.

-> We have removed this part from the main text and moved it to the first page of submission.

Best regards,
Richard Schuster

Research Associate, Carleton University
Email: richard.schuster@glel.carleton.ca

Reviewer #2 (Remarks to the Author):

I found the MS entitled “Effectiveness of protected areas in conserving tropical forest birds” to be highly interesting, important, innovative, and well written. I have but few suggestions that could potentially help cement the authors claims regarding the effectiveness of PAs in protecting vulnerable bird species.

-> We thank Reviewer 2 for their positive comments and constructive suggestions.

Major comments

The authors claim that PAs were established in more remote, and less desirable locations to begin with. However, there is also a possibility that these regions were specifically chosen to protect at risk (bird) species. I think that this option should at least be raised. Furthermore, there could potentially be some more test to try to disentangle this chicken and egg conundrum.

-> Chicken and egg conundrum is a very good description. It is well established that PA location is biased towards more remote and less desirable locations (e.g., high altitude, low productivity areas; e.g., Joppa and Pfaff 2009 in Plos One; Venter et al. 2018 in Conservation Biology), which may create a bias against high biodiversity areas. In parallel, PAs are often established in (some of) the best remaining habitat patches, which may create a bias towards high biodiversity areas. The two are not necessarily contradictory: the best remaining habitat patches at the time of PA creation may be in remote locations that were not particularly biodiversity rich to start with. We have spent a good time thinking about how to separate the two effects, which we discuss in detail in Supplementary Methods 4D. As explained, we believe we have adequately controlled for biases in PA location. We now refer explicitly to this in the main text to ensure readers do not miss it.

First, I think that the year PAs were established should be used as another predictor in some of the tests – to see if PAs established earlier are doing better than those more recently established. These data should be available from the UNEP-WCMC database.

This is a very good suggestion, even though we note two caveats. First, while the WDPA indeed includes a date for each PA (field status_yr), it does not necessarily correspond to the year that given territory was first protected, but to the year of establishment of the current PA (e.g., if a Game Reserve designated in 1990 changed status to National Park in 2005, the status year for the National Park designation will be 2005 and the earlier Game Reserve will no longer be in the WDPA), which may mask a potential increase in PA effectiveness with PA age. Second, whereas earlier PAs were more frequently established to protect scenic landscapes or particular resources (e.g. game), recent decisions on PA location are more likely to have incorporated better data on the distribution of, and threats to, biodiversity, including a stronger focus on threatened species (even because much of those data are themselves quite recent), and so PAs are not necessarily expected to have had less impact over time. With this in mind, we agree that it is interesting to test whether there is a relationship between PA age and effectiveness, given that a negative trend (i.e., if older PAs have higher effectiveness) would reinforce our assumption that the differences in terms of bird biodiversity measured in our study can be interpreted as measures of PA effectiveness. We have thus modelled, for each model used in Analysis I (8 hotspots × 4 bird indices), the link between residuals

(i.e., the remaining difference in bird diversity indices that is not explained by protection, duration, expertise, latitude, longitude, remoteness, altitude, agricultural suitability) and PAs status year. Most (19 out of 32) of these relationships are not significant, but those that are mainly go in the expected direction.

Given space constraints and the nuances needed to interpret this result (the two caveats above) we have opted to present it in supplementary materials rather than in the main text. We have added a phrase in the main text to point to this additional analysis: “We have also found that older protected areas tend to be more effective (in terms of conserving bird diversity; analysis I), consistent with a cumulative implementation effect of protected areas (Supplementary Methods 4E, Extended Fig. 18).”

Furthermore, beyond the metrics of forest quality explored here, I would suggest looking at the Hansen forest cover-change layer (<https://earthenginepartners.appspot.com/science-2013-global-forest>) which gives nearly 20 years spatial data regarding forest loss. There are similar time-series derived spatial datasets spanning even longer periods. Here too the dynamics of the forest cover/loss are important for linking longer-term biodiversity phenomena.

-> This is a very useful suggestion and we thank the reviewer for it. We have followed this proposal by adding a new analysis that investigates how deforestation rates differ inside protected areas versus counterfactual unprotected sites. Our finding that on average PAs experienced 46.7% lower deforestation rates (Extended Table 1) is consistent with the interpretation that our measure of effectiveness reflects recent trends in habitat rather than only biases in the original protected areas locations.

Put together, placing your results within such an (also) temporal perspective should give them even stronger foundations.

-> These two analyses have indeed given stronger foundations to our results!

Minor comments

Could you provide in the supp. the power of your different tests for different groups of species, or regions. This could help compare significant / non-significant results across your many tests with different sample sizes.

-> We have added a table (Extended Table 3) with the number of checklists (so the number of samples used in each test), P-values, and R-squared to give more insights on statistical power.

Could you please also provide a supp. with the lists of species belonging to different categories in different regions, this should help reproducibility of your results, and further explorations of them.

-> This table has been added as a spreadsheet in this current submission. Thank you for this great suggestion.

Please note that the Simrad et al. 2011 data is limited to a max canopy height of 40m. I suggest at least mentioning this.

-> Indeed, we forgot to specify this. We have now added this information in the Supplementary Methods: “limited to maximum canopy heights of 40m”

Reviewers' Comments:

Reviewer #2:

Remarks to the Author:

Overall I think this MS is improved and should be ready for publication following minor changes/addition

I think it is worth mentioning somewhere in your main text clearly that your measures of bird diversity which are not species richness are corrected for species richness – i.e. values of first specialists, endemics. Or threatened (and NT) above and beyond what is expected by richness. In most cases richness is a linear positive predictor of the other measures.

Font in lines 407-408 should be regular text font and not notation font.

I'd suggest including all the references mentioned in the supp, in the supp reference list (even if they appear in the main text) to aid the readers.

Did you try other values for your smoothing parameter (k) other than 4 in your GAM models, to see if the patterns remain?

Also, how important was your control parameter that included the lat & long, I suspect that this parameter would contain much of the variation across sites.

REVIEWERS' COMMENTS:

Reviewer #2 (Remarks to the Author):

Remarks to the Author:

Overall I think this MS is improved and should be ready for publication following minor changes/addition

-> We would like to thank Reviewer 2 for their careful look at our manuscript revision.

I think it is worth mentioning somewhere in your main text clearly that your measures of bird diversity which are not species richness are corrected for species richness – i.e. values of first specialists, endemics. Or threatened (and NT) above and beyond what is expected by richness. In most cases richness is a linear positive predictor of the other measures.

-> We agree with the reviewer and have highlighted this in the main text: “Indeed, **controlling for overall richness**, we find for each of these three groups significant positive effects of protected areas across hotspots”

Font in lines 407-408 should be regular text font and not notation font.

-> We have applied this change

I'd suggest including all the references mentioned in the supp, in the supp reference list (even if they appear in the main text) to aid the readers.

-> We know provide a list of reference at the end of the unique PDF supplementary file, with all references used across the SI.

Did you try other values for your smoothing parameter (k) other than 4 in your GAM models, to see if the patterns remain?

-> Patterns are robust to changes in the smoothing parameter but curves gain in complexity. We considered that $k=4$ was the most consistent value in ecological terms. We have specified in Supplementary methods: “Results were robust to changes in the degree of smoothing function”

Also, how important was your control parameter that included the lat & long, I suspect that this parameter would contain much of the variation across sites.

-> Our control for latitude and longitude carries some information but is not the major factor explaining bird diversity (often duration, expertise and altitude are responsible for a larger amplitude of diversity as can be seen in Supplementary Figures 10-17). Regardless of the amplitude of the impact, it is unlikely to bias our results as our control applies at scale much larger than protected areas (several hundreds of kilometres) as can be seen in Supplementary Figures 10-17. Moreover, our results are conservative towards such bias (i.e., our control could shadow some covariates effects but not create an artefact effect of protected areas).